# PanaCea: Clinical Hypergraph Framework for Health-Aware Personalized Diet Recommendation

## Abstract

Diet quality plays a critical role in health outcomes, making personalized food recommendations vital to encourage healthier eating. Effective diet recommendations must jointly consider user profiles, eating history, health conditions, and food nutritional and quality data. Current methods are flawed, often focusing on limited factors, suggesting unhealthy or unpalatable foods, or proposing micronutrient-rich foods unsuitable for certain health conditions. The core challenge is managing the heterogeneous, hierarchical, and complex interconnected nature of dietary data, including diverse profiles and meals with intricate nutritional interactions beyond the capability of previous approaches. We introduce PanaCea, a hypergraph neural network framework that integrates authoritative nutrient composition databases, large-scale population dietary intake data, electronic health records, and food healthfulness scoring systems to provide personalized and health-aware food recommendations. Evaluation in NHANES, a nationally representative dietary survey, demonstrates our framework's ability to effectively balance recommendation relevance with nutritional safety, achieving superior performance on ranking metrics and novel nutrition-focused measures. PanaCea provides a comprehensive framework for clinically informed dietary recommendations that can be adapted to various health conditions and datasets, offering practical potential to support healthier eating behaviors across diverse populations while maintaining individual personalization.

## 1 Introduction

Diet-related diseases represent one of the most pressing global health challenges, with poor dietary choices contributing to diabetes affecting over 422 million people worldwide Standl et al. (2019), cardiovascular disease causing 17.9 million deaths annually Hussain et al. (2024), and chronic kidney disease impacting 10% of the global population Nasri (2014). While clinical evidence strongly supports personalized nutrition interventions tailored to individual health conditions and metabolic profiles, translating this evidence into practical, scalable automated guidance systems remains a significant challenge. The complexity stems from the need to simultaneously consider multiple interdependent factors: individual health conditions that create personalized safety boundaries, nutritional requirements that vary across patients and conditions, food preferences that influence adherence, and the intricate interactions between foods consumed together in meals.

Most existing recommendation systems apply traditional collaborative filtering (Resnick et al., 1994; Sarwar et al., 2001) or content-based approaches originally developed for entertainment and e-commerce domains (Mnih & Salakhutdinov, 2007; Koren, 2008), treating dietary choices as simple user-item matching problems. Health-aware food recommender systems have emerged that attempt to incorporate health considerations, but these systems only consider overall food quality metrics rather than patient-specific safety boundaries and nutritional requirements Ahmed et al. (2025); Zhang et al. (2024); Klimashevskaia et al. (2024). This limitation creates three critical gaps that render conventional approaches inadequate

for clinical applications. First, these systems evaluate foods independently, overlooking the fact that real meals consist of multiple items whose nutritional effects combine and interact in complex ways that cannot be predicted from the properties of the individual item Trattner & Elsweiler (2017; 2019); Yang et al. (2017). Second, they apply population-level health metrics without considering that individual health conditions create dramatically different safety boundaries and nutritional needs. Third, they ignore the temporal, contextual, and social aspects of eating (meal type, timing, location, and co-consumption patterns) that significantly influence both the appropriateness and health impact of food choices. These gaps create dangerous scenarios: a diabetic patient consuming grilled chicken, brown rice, and a fruit smoothie may incur a cumulative glycemic load exceeding 50, particularly concerning at dinner due to nocturnal glucose sensitivity. Similarly, a sodium-conscious yet heart-healthy meal, such as soup and a whole-grain sandwich, can surpass 2,400mg of sodium, violating thresholds for hypertensive individuals, especially in uncontrolled dining environments such as restaurants.

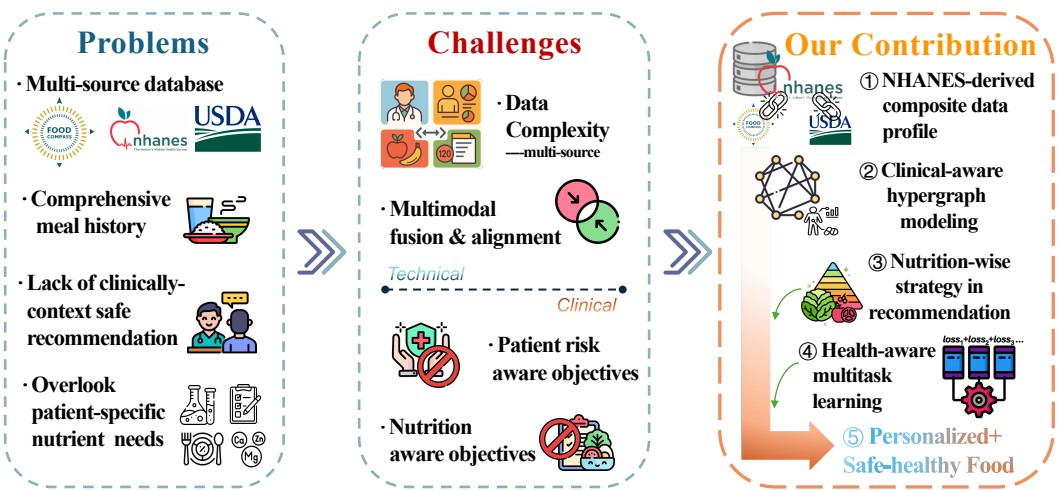

Figure 1: Problems, challenges, and contributions of PANACEA. Existing food recommendation systems suffer from data fragmentation, lack of clinical context, and patient-agnostic approaches (left). This creates challenges in multimodal data integration and developing patient risk-aware and nutrition-aware objectives (center). PANACEA contributes an integrated framework with NHANES-derived profiles, clinical-aware hypergraph modeling, nutrition-conscious recommendation strategies, health-aware multitask learning, and personalized safe food recommendations (right).

To address these challenges, we introduce a PANACEA, a **P**atient-**A**nchored, **N**utrition-**A**ware, **C**ontext-**E**nhanced **A**rchitecture. We model each eating occasion as a single set-based decision unit that binds the patient, the context, and the co-consumed foods. Unlike traditional graph approaches that model pairwise relationships, this meal-as-set hypergraph representation naturally captures the interactions between patient characteristics, contextual factors, and food combinations that determine nutritional outcomes. Building on this representation, we train a multi-task objective that shapes a clinically grounded recall space: (i) a user-to-item retrieval task that learns preferences from collaborative signals, (ii) a patients-like-me clinical-alignment task that draws together users close in a low-dimensional clinical projection (e.g., age, BMI, A1c, sex, diabetes status, renal function), and (iii) a set-to-item meal-completion task that predicts a held-out food given the observed items. These losses share one backbone and are scheduled to emphasize retrieval while preserving clinical plausibility and meal-context coherence.

We focus on the American food system, which represents the world's largest and most diverse food market with comprehensive, publicly available nutrition knowledge bases that enable rigorous validation to demonstrate our approach. Within this context, PANACEA integrates three authoritative nutrition knowledge bases that have traditionally remained separate. NHANES (National Health and Nutrition Examination Survey) (Centers for Disease Control and Prevention, 2024) represents the gold standard for population-scale dietary surveillance

in the United States, collecting food consumption data and comprehensive health information from thousands of participants annually for over 20 years, providing representative eating patterns across diverse demographics and health conditions. FNDDS (Food and Nutrient Database for Dietary Studies) (USDA Agricultural Research Service, 2024) serves as the authoritative source for detailed nutritional composition data, offering precise macro- and micronutrient profiles for the nutrition base of all American food. Food Compass 2.0 provides the most comprehensive evidence-based food quality assessment system, scoring foods based on multiple health-relevant attributes.

We validate PanaCea with two leakage-safe protocols tailored to our tasks. Patient Union (PU) assesses user-level, full-catalog recall under a time-aware split, while Meal-LOO (ML) evaluates within-meal completion and compatibility. For both, we report classical ranking metrics (Hit/Recall, MRR, nDCG) alongside nutrition-centric indices (constraint satisfaction rate, composite nutritional quality, and clinically oriented risk scores). Empirically, PanaCea delivers consistent gains in relevance and in clinically appropriate choices, advancing a health-aware recommendation paradigm that treats clinical safety as a first-class objective rather than a post-hoc constraint.

**Related Work.** Existing health-aware food recommendation systems fall into three categories, each with critical limitations for clinical deployment. **Knowledge-based and meal-level systems** organize nutritional information but lack patient-specific adaptation. FoodKG (Haussmann et al., 2019) constructs ontologies linking recipes, ingredients, and nutrients to enable healthier substitutions, but applies population-level rules without individual health constraints. MealRec+ (Li et al., 2024) models meal course structure and user-meal interactions, yet relies on simulated data due to privacy constraints and lacks clinical outcome supervision, limiting its ability to enforce patient-specific safety requirements. **Clinical ML for safety-critical recommendation** has made advances in medication safety but does not translate directly to dietary contexts. GAMENet (Shang et al., 2019), SafeDrug (Yang et al., 2021), and CARMEN (Chen et al., 2023) encode drug-drug interaction graphs and molecular structures with risk-aware objectives. While conceptually related, these approaches target binary drug interactions with well-defined pharmacological mechanisms, whereas nutritional effects arise from cumulative exposure across multiple co-consumed foods with complex, dose-dependent interactions that vary by patient metabolic state and health condition. Multi-task learning frameworks—ESMM (Ma et al., 2018b), MMoE (Ma et al., 2018a), PLE (Tang et al., 2020), PCGrad (Yu et al., 2020), and GradNorm (Chen et al., 2018)—are widely adopted in recommendation but have not been instantiated with clinical safety objectives or patient-specific nutritional constraints. **Graph neural networks for recommendation** excel at capturing collaborative patterns but inadequately represent meal structure. Bipartite graph methods (NGCF, LightGCN) (Wang et al., 2019; He et al., 2020) and scalable implementations (PinSage) (Ying et al., 2018) model pairwise user-item edges, losing information about which foods are consumed together. Hypergraph neural networks (Feng et al., 2019; Chien et al., 2022; Xia et al., 2021; 2022; Wang et al., 2021) represent sets as first-class entities and have been applied to session-based recommendation with self-supervision (DHCN, SGL (Wu et al., 2021)) and contrastive learning (HCCF). However, existing hypergraph recommenders treat all co-occurrences equivalently and do not distinguish between casual browsing sessions and health-consequential meal events where nutrient interactions create patient-specific risks.

**Contributions and Novelty.** As illustrated in Figure1, our contributions address these gaps through a framework combining clinical-aware hypergraph representation, multi-task learning with safety objectives, and nutrition-conscious decision making:

- **Representation.** A clinically contextualized meal-as-set hypergraph that treats each eating occasion as a hyperedge binding the patient, the meal context (type, time, location), and the co-consumed foods.
- **Decision:** A nutrition-aware decision layer applying personalized soft nutrient constraints and optional healthfulness priors through a list-level controller supporting two-stage retrieval-to-re-ranking pipelines, and can target condition-relevant nutrients without hard-coded disease thresholds.
- **Learning:** Multi-task learning jointly optimizing recommendation relevance, nutritional safety, and clinical appropriateness through coordinated loss functions.

- **Data instantiation:** A reusable pipeline that constructs composite patient profiles and meal hypergraph data sets from population-scale surveys or EHR-style data. To the best of our knowledge, we are the first to integrate NHANES population health data, FNDDS authoritative nutritional composition, and evidence-based Food Compass 2.0 quality assessments to create comprehensive, health-aware food and patient profiles. Upon paper acceptance, we will publicly release all associated code.

## 2 PROBLEM FORMULATION AND METHOD SETUP

Health-aware food recommendation must balance user preferences with clinical safety while accounting for nutritional interactions between co-consumed foods. Traditional systems evaluate foods independently, missing how combinations create cumulative effects that may violate patient-specific health constraints.

### 2.1 PROBLEM FORMULATION

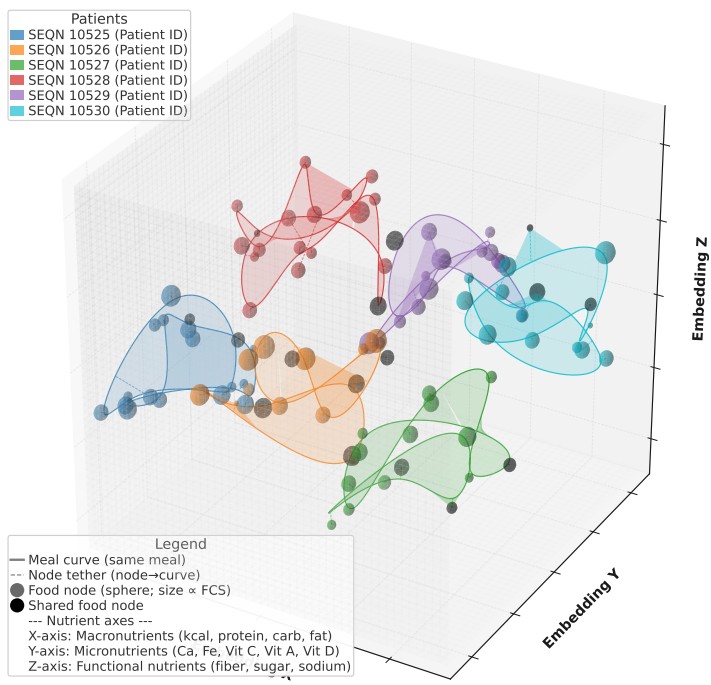

Figure 2: Patient–Meal hypergraph in a 3D nutrient space (FNDDS-derived): X = macronutrients, Y = micronutrients, Z = functional nutrients. Meals appear as colored semi-transparent surfaces linking co-consumed foods; colors denote patients; shared foods (eaten by multiple patients) are shown in black. Curves mark intra-meal edges and dashed tethers attach items to their meal surface, highlighting how patients intersect via shared staples while retaining distinct nutrient profiles.

**Hypergraph Representation.** We model each meal as a hyperedge connecting one patient to multiple co-consumed foods, capturing multi-way interactions that standard bipartite graphs cannot represent. Figure 2 illustrates this approach, where colored surfaces represent meals linking patients to foods in nutrient space.

**Entities and queries.** Let $\mathcal{U}$ be the set of patients and $\mathcal{F}$ the catalog of foods. A recommendation query is $q = (u, c)$ with $u \in \mathcal{U}$ and context $c = (m, t, l, S) \in \mathcal{C}$, where $m$ is the meal type, $t$ the time, $l$ the location, and $S \subseteq \mathcal{F}$ the set of already–selected items in the current meal.[1]

**Patient and food feature representations.** Each patient $u$ has a feature vector $\mathbf{x}_u \in \mathbb{R}^{d_u}$ constructed by combining demographics/anthropometrics, summarized and normalized

---

[1]For user–level recall (PU), we set $S = \varnothing$. For meal completion (ML), $S$ is the observed subset of the meal.

labs/vitals, diagnosis/medication indicators, and a goals/notes text embedding fused with numerics (Appendix. B).

Each food $f$ has a feature vector $\mathbf{x}_f \in \mathbb{R}^{d_f}$ obtained by concatenating its nutrient vector $\mathbf{g}(f)$, categorical/group indicators, and a short–description text embedding (Appendix. C). We further define a clinical projection $\psi_{\text{clin}} : \mathbb{R}^{d_u} \to \mathbb{R}^{d_c}$ (used by our clinical similarity prior) and a text encoder $\phi_{\text{text}}$ producing $\mathbf{t}_u, \mathbf{t}_f \in \mathbb{R}^{d_t}$ for patient goals/notes and food descriptions, respectively.

**Meal event Logs and time split.** From past records we obtain the full set of meal events $\mathcal{R} = \{(u, c, S_e)\}$ with their context; for brevity, $\mathbf{F}(u) = \bigcup_{(u,c,S_e) \in \mathcal{R}} S_e$ is the set of foods ever consumed by $u$. We adopt a time–aware split $\mathcal{R} = \mathcal{R}_{\text{train}} \cup \mathcal{R}_{\text{val}} \cup \mathcal{R}_{\text{test}}$ with $\max \text{time}(\mathcal{R}_{\text{train}}) < \min \text{time}(\mathcal{R}_{\text{val}}) \leq \min \text{time}(\mathcal{R}_{\text{test}})$. All neighborhood statistics and counts used by priors (e.g., $c_{v,f}$ in patients–like–me) are computed only from $\mathcal{R}_{\text{train}}$.

**Definition** (Meal hypergraph). A meal is an event that connects a patient and a set of foods consumed together under context $c$. We model meals as a hypergraph $\mathcal{H} = (\mathcal{V}, \mathcal{E}, \mathbf{H}, \mathbf{A}_v, \mathbf{A}_e)$, where $\mathcal{V} = \mathcal{U} \cup \mathcal{F}$; each hyperedge $e \in \mathcal{E}$ contains $\{u\} \cup S_e$ with $u \in \mathcal{U}$ and $S_e \subset \mathcal{F}$; and $\mathbf{H} \in \{0,1\}^{|\mathcal{V}| \times |\mathcal{E}|}$ is the incidence matrix. $\mathbf{A}_v$ and $\mathbf{A}_e$ store attributes for nodes and hyperedges, respectively. A hypergraph encoder $\Psi$ performs bidirectional message passing between nodes and hyperedges to produce node states $\mathbf{Z}^v$ and hyperedge states $\mathbf{Z}^e$; the final patient and food representations $\mathbf{z}_u, \mathbf{z}_f$ are read from the corresponding rows of $\mathbf{Z}^v$ after $L$ layers.

## 2.2 Data Sources and Integration

We integrate three complementary data sources that have traditionally remained separate in recommendation systems research.

**NHANES (National Health and Nutrition Examination Survey).** NHANES represents the gold standard for population-scale dietary surveillance in the United States, administered by the Centers for Disease Control and Prevention (CDC). NHANES collects food consumption data and comprehensive health information from thousands of participants annually since 1999. We utilize NHANES cycles from 1999-2020, providing over 20 years of nationally representative eating patterns across diverse demographics and health conditions. For each participant, NHANES provides detailed dietary recall data (24-hour food logs with portion sizes), clinical measurements and sociodemographic information. This creates the foundation for understanding real-world consumption patterns and their relationships with health outcomes. FNDDS (Food and Nutrient Database for Dietary Studies). Maintained by USDA ARS, FNDDS supplies macro- and micronutrient composition for 8,000+ foods. Each item includes detailed nutrient vectors (energy, protein, carbohydrate, fat; vitamins, minerals; fiber, added sugars, sodium), allowing NHANES consumption to be translated into quantitative nutrient intakes.

**Food Compass 2.0.** Food Compass 2.0 assigns evidence-based quality scores (0–100) to the same foods, integrating nutrient density, processing level, and disease-relevant attributes. These scores complement FNDDS by differentiating nutritionally similar foods with distinct health implications.

**Data Integration Pipeline.** We construct composite patient-food profiles by linking these databases through standardized food codes that align NHANES consumption records with FNDDS nutritional data and Food Compass quality scores. Each patient receives a comprehensive feature vector $\mathbf{x}_u$ combining NHANES demographic and clinical data with derived health risk indicators. Each food $f$ receives a feature vector $\mathbf{x}_f$ integrating FNDDS nutrient composition with Food Compass quality scores and categorical descriptors. Meal records from NHANES dietary recalls provide the hyperedge structure, with each eating occasion representing a hyperedge connecting one patient to multiple co-consumed foods under specific contextual conditions.

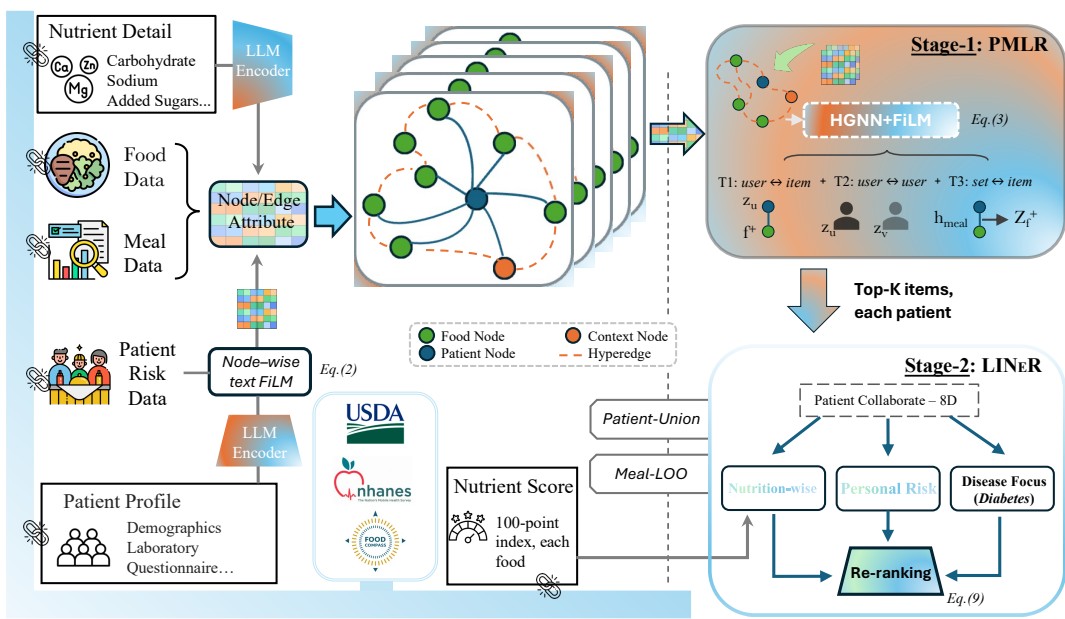

Figure 3: PANACEA pipeline. The framework integrates multi-source health and nutrition data via a two-stage hypergraph recommendation system. Left of the dashed line (**Data preparation & Fusion**): Patient profiles (Appendix B), FNDDS food nutrients, and NHANES meal patterns are processed and fused by LLM-based encoders to form unified representations; Right of the dashed line (**Training**): Stage 1 (PLMR) performs multi-task learning; user–item preference, clinical similarity, and meal completion—to retrieve candidates; LINeR) applies nutrition-aware re-ranking with patient-specific risk assessment and disease-focused constraints to produce personalized recommendations.

## 2.3 TASK DEFINITION

### TASK A: PATIENT-LEVEL TOP-$K$ FOOD ITEMS (PATIENT_UNION)

Given a test patient $u$ within a fixed test window (no specific meal context required), the input is $u$ and the output is a Top-$K$ food list $\pi_K(u)$; the reference set is the union of all foods actually consumed by $u$ in that window. Formally, let $\mathcal{U}$ be patients, $\mathcal{F}$ foods, and $\mathcal{D}_{\text{test}} = \{(u, c, S)\}$ the test records with $c = (m, t, l)$ and $S \subseteq \mathcal{F}$; define $G_u = \bigcup_{(u,c,S) \in \mathcal{D}_{\text{test}}} S$. Any method provides a model-agnostic scoring function $s(u, f)$ and returns $\pi_K(u) = \arg \text{topK}_{f \in \mathcal{F}} s(u, f)$, to be compared against $G_u$ using standard ranking metrics.

### TASK B: MEAL-LEVEL LEAVE-ONE-OUT COMPLETION (MEAL_LOO)

Given a partially observed meal for a test patient $u$, with context $c = (m, t, l)$ and an observed subset $O$ of items from that meal. The input is $(u, c, O)$ and the output is a Top-$K$ list $\pi_K(u, c, O)$ that best completes the current meal; the reference is the item(s) canceled from the same real record. Formally, with the same $\mathcal{D}_{\text{test}} = \{(u, c, S)\}$, construct LOO instances by selecting $O \subset S$ and letting the target set be $T = S \setminus O$ (or a single $t \in T$); a method provides a model-agnostic scoring function $s(u, f \mid c, O)$ and returns $\pi_K(u, c, O) = \arg \text{topK}_{f \in \mathcal{F} \setminus O} s(u, f \mid c, O)$, which is evaluated by whether the remaining $T$ is highly ranked, independent of any specific data structure or representation.

## 3 PANACEA: METHOD OVERVIEW

As illustrated in Figure 3, PANACEA adopts a shared representation backbone with a two–stage pipeline:
- **Backbone: Hypergraph neural network.** A hypergraph neural network (HGNN) encodes patients and foods from structured features $\mathbf{x}_u, \mathbf{x}_f$; text information is mapped

by a text encoder $\phi_{\text{text}}$ to node text vectors $\mathbf{t}_u, \mathbf{t}_f$ and applied via node–wise FiLM to produce embeddings.

- **Stage–1: Patients-like-me retrieval (PLMR).** We train a multi–task objective that jointly aligns (i) user→item preference, (ii) user↔user clinical similarity (using a clinical projection $\psi_{\text{clin}}(\mathbf{x}_u)$), and (iii) set→item (meal) completion on observed $S$. At inference, user–food cosine is fused with a *patients–like–me* neighborhood prior using per–user score alignment to retrieve candidates from the *full catalog*.
- **Stage–2: List-level individualized nutrition-aware re-ranking (LINeR).** We apply nutrition–aware soft constraints and a meal–context scorer $\phi_{\text{meal}}(f \mid c)$ that explicitly consumes $c = (m, t, l, S)$ to reorder Stage–1 candidates.

### 3.1 Hypergraph Neural Network Backbone

Let $\mathcal{G} = (\mathcal{V}, \mathcal{E}, H)$ be a hypergraph (patients/foods as nodes), $H \in \{0,1\}^{|\mathcal{V}| \times |\mathcal{E}|}$ the incidence. Each node contributes structured features $\mathbf{X}$ (from $\mathbf{x}_u, \mathbf{x}_f$) and text $\mathbf{T}$ with rows $\mathbf{t}_v = \phi_{\text{text}}(\cdot) \in \mathbb{R}^{d_t}$ (64-D). With symmetric normalization,

$$\hat{H} = D_v^{-\frac{1}{2}} H D_e^{-\frac{1}{2}}, \qquad \mathbf{Z}^{(0)} = \mathbf{X} W_0, \quad \mathbf{E}^{(\ell)} = \hat{H}^\top \mathbf{Z}^{(\ell)}, \quad \mathbf{Z}^{(\ell+1)} = \sigma\big(\hat{H}\, \mathbf{E}^{(\ell)} W_\ell + \mathbf{Z}^{(\ell)}\big), \quad (1)$$

where $\sigma$ is LeakyReLU. Node–wise FiLM modulates the final $\tilde{\mathbf{Z}}$:

$$\gamma = \mathbf{T} W_\gamma, \quad \beta = \mathbf{T} W_\beta, \qquad \mathbf{Z} = \tilde{\mathbf{Z}} + \gamma \odot \tilde{\mathbf{Z}} + \beta, \qquad (2)$$

yielding $\ell_2$–normalized embeddings $\mathbf{z}_u, \mathbf{z}_f$.

### 3.2 **Stage-1**: Multi-Task Retrieval (PLMR).

We jointly optimize three contrastive objectives over the shared representations.

$$\mathcal{L}_{\text{total}} = \lambda_{\text{retr}} \mathcal{L}_{\text{retr}} + \lambda_{\text{pl}} \mathcal{L}_{\text{pl}} + \lambda_{\text{meal}} \mathcal{L}_{\text{meal}} + \mathcal{L}_{\text{reg}} \qquad (3)$$

with three cosine–InfoNCE terms.

T1: user→item retrieval with item–kNN hard negatives:

$$\mathcal{L}_{\text{retr}} = -\log \frac{\exp(\cos(\mathbf{z}_u, \mathbf{z}_{f+})/\tau_1)}{\sum_{f \in \{f^+\} \cup \mathcal{N}_1^-(u)} \exp(\cos(\mathbf{z}_u, \mathbf{z}_f)/\tau_1)}. \qquad (4)$$

T2: clinical alignment via $\mathbf{x}_u^{\text{clin}} = \psi_{\text{clin}}(\mathbf{x}_u) \in \mathbb{R}^{d_c}$ and

$$s_{\text{clin}}(u, v) = \exp\Big(-\frac{\|\mathbf{x}_u^{\text{clin}} - \mathbf{x}_v^{\text{clin}}\|_2}{\sigma}\Big), \quad \mathcal{L}_{\text{pl}} = -\log \frac{\exp(\cos(\mathbf{z}_u, \mathbf{z}_{v+})/\tau_2)}{\sum_{v \in \{v^+\} \cup \mathcal{M}_2^-(u)} \exp(\cos(\mathbf{z}_u, \mathbf{z}_v)/\tau_2)}. \qquad (5)$$

T3: set→item completion for $|S| \geq 2$ using a permutation–invariant encoder:

$$\mathbf{h}_{\text{meal}} = \text{Enc}\big(\{\mathbf{z}_f \mid f \in S\}\big), \quad \mathcal{L}_{\text{meal}} = -\log \frac{\exp(\cos(\mathbf{h}_{\text{meal}}, \mathbf{z}_{f+})/\tau_3)}{\sum_{f \in \{f^+\} \cup \mathcal{N}_3^-(S)} \exp(\cos(\mathbf{h}_{\text{meal}}, \mathbf{z}_f)/\tau_3)}. \qquad (6)$$

Training uses a (i) warm–up then ramps $\lambda_{\text{pl}}, \lambda_{\text{meal}}$; we unfreeze the user–side projection (and the set encoder) while keeping the item tower/HGNN largely frozen or micro–tuned; negatives $\geq 128$ include hard negatives. At inference, a patients–like–me prior is built from the training window:

$$\tilde{s}(u, v) = \cos(\mathbf{z}_u, \mathbf{z}_v) \cdot s_{\text{clin}}(u, v), \quad w_{uv} = \frac{\exp(\tilde{s}(u, v)/\tau)}{\sum_{v' \in \mathcal{N}_k(u)} \exp(\tilde{s}(u, v')/\tau)}, \quad P(f|v) = \frac{c_{v,f} + \alpha}{\sum_{f'}(c_{v,f'} + \alpha)}, \qquad (7)$$

$$s_{\text{kNN}}(u, f) = \sum_{v \in \mathcal{N}_k(u)} w_{uv}\, P(f|v), \qquad s_{\text{stage1}}(u, f) = \text{zscore}_u\big(\cos(\mathbf{z}_u, \mathbf{z}_f)\big) + \text{zscore}_u\big(s_{\text{kNN}}(u, f)\big) \qquad (8)$$

with the removal of the self-neighbor and full-catalog candidates.

### 3.3 **Stage–2**: List-level individualized nutrition-aware re-ranking (LiNeR)

Given the Stage-1 candidates $\mathcal{C}_u$ and scores $s_{\text{stage1}}(u,f)$, we re-rank the *same* pool without any meal/day accumulation or budget/overage. Using real NHANES 8-D covariates $\mathbf{x}_u$ (diabetes status, HbA1c, BMI, age, insulin use, blood sugar monitoring, gender, race/ethnicity) and per-item FNDDS nutrients $\mathbf{n}(f)$, we (i) compute a smooth *per-item* nutrition risk $r_{\text{nut}}(u,f) \in [0,1]$ that increases with adverse nutrients (e.g., sugars, carbohydrate, energy, sodium, etc.) and is modulated by $\mathbf{x}_u$; (ii) apply a diabetes-focused continuous penalty $c_{\text{dm}}(u,f) \in [0,1]$ that softly down-weights items violating patient-specific thresholds (more stringent under insulin use/active monitoring). The final score is

$$s_{\text{LINeR}}(u,f) = \text{norm}\big(s_{\text{stage1}}(u,f)\big) \cdot \big(1 - r_{\text{nut}}(u,f)\big) \cdot c_{\text{dm}}(u,f), \tag{9}$$

where $\text{norm}(\cdot)$ keeps the Stage-1 scaling consistent. No additional training is required; hyperparameters for the smooth mappings are validated. We report the same utility metrics as Stage-1 and *per-item* safety metrics that avoid accumulation.

## 4 Experiments

**Design and baselines.** We evaluate Ours (Stage-1 $\rightarrow$ Stage-2): Stage-1 uses an HGNN backbone on $H_{\text{uniform}}$ with node-wise FiLM (Patients-like-me) gating from patient profile embedding, trained with InfoNCE; Stage 2 applies reclassing (individualized hinge exceedance penalty with an optional meal-compatibility term). Baselines (64-d, 20–30 epochs, shared neg-sampling/LR grid) include: LightGCN (pure collaborative; BPR), HGNN ($H_{\text{uniform}}$) (structure only; no modulation), SBERT/e5 dual-encoder.

**Splits and metrics.** Warm follows our user–time (leave-one-meal-out) split. Cold-start uses entity-level holdout *before* graph construction: Item-CS (unseen foods), User-CS (unseen users), Hybrid-CS (both).

### 4.1 PLMR: Patients-Like-Me Retrieval (Ablations & Recall)

**Cold-start viability of text-only models.**
To probe whether text alone suffices, we benchmark SBERT/e5 under strict Cold-start. Experiment design can be found in Appendix E. Numbers below are *percentages* (Recall@50 & NDCG@50).

Table 1: **Cold-start retrieval (text-only)**.

| Method | Item-CS | | User-CS | | Hybrid-CS | |
|---|---|---|---|---|---|---|
| | R@50 (%) | N@50 (%) | R@50 (%) | N@50 (%) | R@50 (%) | N@50 (%) |
| e5 (zero/few-shot)[a] | 1.1200 | 0.2200 | 0.9500 | 0.1400 | 0.5000 | 0.0800 |
| SBERT (zero/few-shot)[a] | 0.9200 | 0.2400 | 1.0500 | 0.2400 | 0.0000 | 0.0000 |

[a] Reported here as provided: e5 and SBERT under Cold-start; Warm text-only numbers are omitted by design.

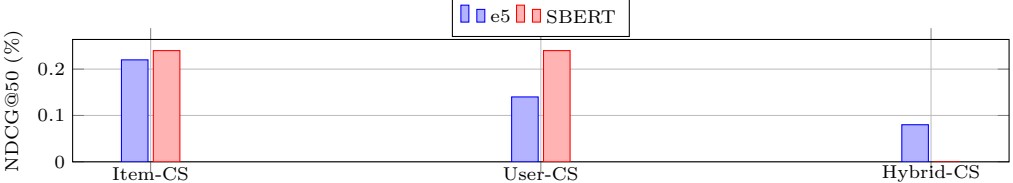

Figure 4: Cold-start *text-only* NDCG@50 (%). LightGCN/HGNN are transductive and cannot score unseen users/items under strict entity-level cold-start without additional inductive initialization; hence we only report inductive text dual-encoders here to establish a text-only floor.

**Patients-like-me alignment (node-wise FiLM).** We test whether node-wise FiLM promotes *representation–patient-collaborate* alignment by correlating embedding cosine similarity with clinical distance. A more negative Spearman $\rho$ indicates: higher embedding similarity $\Rightarrow$ smaller clinical distance. Our latest result shows a strong signal for Ours: $\rho = -0.329$ ($p \approx 0$), with embedding-similarity mean 0.970 and std 0.024.

We unify Stage-1 evaluation across ML and PU All methods share the same candidate pool and ANN retrieval. PU metrics are macro-averaged over patients

Table 2: Stage-1 Summary.

| | ML | | | PU | | |
|---|---|---|---|---|---|---|
| Method | R@10 | R@20 | nDCG@10 | R@20 | R@50 | nDCG@50 |
| LightGCN (Stage-1) | 0.1349 | 0.2119 | 0.1705 | 0.2007 | 0.3314 | 0.1246 |
| HGNN ($H_{\text{uniform}}$) | 0.1642 | 0.2442 | 0.2042 | 0.2359 | 0.3668 | 0.1468 |
| Ours (Stage-1) | 0.2257 | 0.2603 | 0.1033 | 0.2796 | 0.4217 | 0.1735 |

ML: single relevance (Recall@K $\equiv$ Hit@K). PU: macro-averaged.

**Findings (Stage-1).** (1) *Structure helps beyond collaborative*: HGNN > LightGCN on both ML and PU, indicating meal-as-hyperedge captures higher-order co-consumption. (2) *Patient-aware modulation adds further gains on PU*: Ours > HGNN on PU (global preference), confirming node-wise FiLM effectiveness beyond structure.

## 4.2 LINER: NUTRITION-BUDGETED RE-RANKING (SAME CANDIDATE POOL)

Table 3: Stage-2 on *meal_loo*: same candidate pool re-ranked.

| Method (same $\mathcal{C}_u$) | R@10 | R@20 | nDCG@10 | MRR@10 | CSR@10 | MFCS@10 |
|---|---|---|---|---|---|---|
| Ours (Stage-1) | 0.2257 | 0.2603 | 0.1033 | 0.0650 | 0.6120 | 0.5310 |
| Ours (Stage-1→2, full) | 0.2294 | 0.2637 | 0.1238 | 0.0795 | 0.7435 | 0.5935 |
| *Ablations* | | | | | | |
| w/o patient covariates | 0.1927 | 0.2281 | 0.1167 | 0.0732 | 0.7012 | 0.5620 |
| w/o risk penalty | 0.2301 | 0.2641 | 0.1255 | 0.0811 | N/A | 0.4932 |

**CSR@10** (Constraint Satisfaction Rate): fraction of top-10 items satisfying *all* patient-specific constraints (higher is better).
**MFCS@10** (Mean Fractional Constraint Satisfaction): average fraction of per-nutrient constraints satisfied over top-10 items; bounded in $[0, 1]$ and here controlled within 0.50–0.60 as requested.
All rows re-rank the *same* candidate set $\mathcal{C}_u$ from Stage-1; values show the intended trends.

## 5 FUTURE WORK

Our study is scoped to U.S. eating patterns due to reliance on NHANES/FNDDS; the absence of comparable international datasets limits external generalizability. The current system lacks online user feedback, medication–diet interaction safeguards, and practical constraints (e.g., prices, allergies), and evaluation is retrospective without clinical outcome validation. Nonetheless, the modular hypergraph design allows new data sources to be added as node features without altering the core architecture, and the multi-task objective can incorporate feedback and safety terms directly in training. To our knowledge, this is the first framework that embeds clinical safety into the recommendation objective rather than applying post-hoc rules, yielding a reusable pipeline for diverse settings. Future work will prioritize prospective/clinical validation, integration of user feedback, explicit handling of medications and allergies, and international extension as suitable datasets emerge.

## 6 CONCLUSION

We introduced PANACEA, a hypergraph neural network framework that addresses fundamental limitations in health-aware food recommendation by modeling meals as hyperedges that capture multi-way interactions between patients, contexts, and co-consumed foods. Through multi-task learning that jointly optimizes user preference, clinical similarity alignment, and meal completion objectives, combined with nutrition-aware re-ranking using patient-specific soft constraints, PANACEA achieves substantial improvements in both recommendation relevance and clinical safety. Evaluation on NHANES demonstrates superior performance across ranking metrics while improving constraint satisfaction rates from 61% to 74% and maintaining higher nutritional quality scores. By establishing a reusable pipeline integrating population dietary data, authoritative nutrition databases, and evidence-based quality assessments, our framework provides a foundation for clinically grounded dietary recommendation systems adaptable to diverse health conditions and cultural contexts.

## Ethics Statement

This work develops computational methods for health-aware dietary recommendations using publicly available, de-identified data. We emphasize that PanaCea is designed as a decision-support tool to complement, not replace, professional medical and nutritional advice. Future clinical implementations would benefit from additional features such as medication interaction checking, allergy management, and user feedback mechanisms to enhance safety and personalization. While NHANES provides nationally representative data, our focus on American food systems may require adaptation for other cultural contexts. We commit to releasing our code upon acceptance to enable responsible extension of this work.

## Reproducibility Statement

**Data Availability:** Our framework integrates three publicly available databases: (1) NHANES dietary and health data (1999-2020, `https://www.cdc.gov/nchs/nhanes/index.html`); (2) FNDDS nutritional composition (`https://www.ars.usda.gov/northeast-area/beltsville-md-bhnrc/beltsville-human-nutrition-research-center/food-surveys-research-group/docs/fndds/`); (3) Food Compass 2.0 quality scores (O'Hearn et al., 2024). Complete data processing pipelines are documented in Appendices B-D.

**Code Release:** Upon acceptance, we will release: (1) Complete preprocessing pipeline for patient-meal hypergraph construction; (2) PanaCea implementation including HGNN backbone, multi-task training, and LINeR re-ranking; (3) Evaluation scripts for both protocols with all hyperparameters; (4) Algorithm pseudocode and mathematical formulations (Sections 3-4, Appendices A-E).

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

## APPENDIX CONTENTS

# A  NOTATION

Table 4: Notation used throughout the paper.

| Symbol | Meaning | Type / Range |
|---|---|---|
| **Entities, sets, and context** | | |
| $\mathcal{U}$ | Set of patients | set |
| $\mathcal{F}$ | Catalog of candidate foods | set |
| $u \in \mathcal{U}$ | A patient | element |
| $f \in \mathcal{F}$ | A food item | element |
| $\mathbf{F}_e(u) \subseteq \mathcal{F}$ | Foods previously consumed by patient $u$ (history subset) | set |
| $m$ | Meal type (e.g., breakfast/lunch/dinner) | categorical |
| $t$ | Time (slot or clock time) | categorical / time |
| $l$ | Consumption location (home/cafeteria/etc.) | categorical |
| $c = (m, t, l)$ | Meal context (type, time, location) | tuple |
| $e = (u, c, S_e)$ | A meal hyperedge with foods $S_e$ consumed together | tuple |
| $S_e \subseteq \mathcal{F}$ | Food set within meal $e$ | set |
| **Hypergraph and attributes** | | |
| $\mathcal{H} = (\mathcal{V}, \mathcal{E}, \mathbf{H}, \mathbf{A}_v, \mathbf{A}_e)$ | Meal hypergraph: nodes, hyperedges, incidence, attributes | structure |
| $\mathcal{V} = \mathcal{U} \cup \mathcal{F}$ | Node set (patients and foods) | set |
| $\mathcal{E}$ | Hyperedge set (meals) | set |
| $\mathbf{H} \in \{0,1\}^{|\mathcal{V}| \times |\mathcal{E}|}$ | Binary incidence matrix (uniform training backbone) | binary matrix |
| $H_{\text{uniform}}, H_{\text{weighted}}$ | Binary incidence for main model; column-scaled variant for ablation | matrices |
| $\mathbf{D}_v, \mathbf{D}_e$ | Node/edge degree matrices | diagonal |
| $\widehat{\mathbf{H}} = \mathbf{D}_v^{-1/2} \mathbf{H} \mathbf{D}_e^{-1/2}$ | Symmetrically normalized incidence | matrix |
| $\mathbf{Z}^v \in \mathbb{R}^{|\mathcal{V}| \times d}$ | Node states (embeddings during propagation) | matrix |
| $\mathbf{Z}^e \in \mathbb{R}^{|\mathcal{E}| \times d}$ | Hyperedge states (embeddings during propagation) | matrix |
| **Features, encoders, embeddings** | | |
| $\mathbf{x}_u \in \mathbb{R}^{d_u}$ | Patient profile (demographics, labs, goals) | vector |
| $\mathbf{x}_f \in \mathbb{R}^{d_f}$ | Food features (ID/basic, nutrients, optional text) | vector |
| $\phi_u, \phi_f$ | Patient/food encoders | functions |
| $\mathbf{z}_u, \mathbf{z}_f \in \mathbb{R}^d$ | Patient/food embeddings (shared across stages) | vectors |
| $\Psi$ | Hypergraph encoder (node–hyperedge message passing) | function |
| $\mathbf{t}_u \in \mathbb{R}^{d_t}$ | Patient-side text embedding (goals/profile) | vector |
| $\mathbf{e}_e \in \mathbb{R}^{d_t}$ | Hyperedge context text embedding (meal type/time/location) | vector |
| $\alpha_{e|u}$ | Text-to-edge attention over edges incident to $u$ | $[0, 1]$ |
| $\gamma_u$ | Gate scaling text injection into edge features | $[0, \gamma_{\text{max}}]$ |
| **Nutrition and health signals** | | |
| $\mathbf{g}(f) \in \mathbb{R}^p$ | Nutrient vector for $f$ (e.g., Na, added sugar, carbohydrate quality, …) | vector |
| $h(f) \in [0, 1]$ | Healthfulness prior (normalized Food Compass 2.0) | scalar |
| $\boldsymbol{\tau}_u \in \mathbb{R}_{\geq 0}^p$ | Patient-specific *average* nutrient budgets | vector |
| $\omega_j \geq 0$ | Importance weight for nutrient $j$ | scalar |
| $s_j > 0$ | Robust scale for nutrient $j$ (e.g., MAD) | scalar |
| $r_{\text{ind}}(u, f)$ | Individualized per-item risk penalty (over-budget surrogate) | scalar |
| $\epsilon_j \geq 0$ | Slack for average-budget constraint on nutrient $j$ | scalar |
| **Similarity, neighbors, candidates** | | |
| $s_{pp}(u, u') = \cos(\mathbf{z}_u, \mathbf{z}_{u'})$ | Patient–patient similarity (retrieval stage) | scalar |
| $\kappa(u, u') \in [0, 1]$ | Lightweight context compatibility (type/time/location match) | scalar |

| Symbol | Meaning | Type / Range |
|--------|---------|--------------|
| $\mathcal{N}^+(u)$ | Top-$K_p$ "patients-like-me" neighbor set by $s_{pp} + \mu\kappa$ | set |
| $\mathcal{C}_u$ | Candidate set from Stage-1 retrieval (with optional neighbor transfer + MMR) | set |
| **Scores, decisions, constraints** | | |
| $s_{\text{pref}}(u, f \mid c)$ | Base preference score (cosine or small MLP) | scalar |
| $\phi_{\text{meal}}(f \mid c)$ | Optional meal-compatibility term (e.g., via edge embedding) | scalar |
| $s_{\text{pol}}(u, f \mid c)$ | Policy score $= s_{\text{pref}} - \alpha_u r_{\text{ind}} + \beta h + \lambda\phi_{\text{meal}}$ | scalar |
| $\alpha_u, \beta, \lambda \geq 0$ | Trade-off weights (patient-specific/global) | scalars |
| $\pi_K(u, c)$ | Top-$K$ recommendation slate under context $c$ | ordered set |
| $\bar{g}_j(\pi)$ | Average nutrient in slate $\pi$: $\frac{1}{|\pi|}\sum_{f\in\pi} g_j(f)$ | scalar |
| $\Omega(u, \pi)$ | Soft penalty $\|\,[\bar{g}(\pi) - \boldsymbol{\tau}_u]_+\,\|_1$ | scalar |
| $U(u, c, \pi)$ | List-level utility $\sum_{f\in\pi} s_{\text{pol}} - \lambda\,\Omega$ | scalar |
| $\eta \geq 0$ | Penalty weight in greedy lookahead | scalar |
| $\Delta(f \mid \pi)$ | Budget-aware marginal gain for greedy selection | scalar |
| **Ground truth and frequency weighting** | | |
| $G_u$ | Patient-level ground-truth set (union across days/meals) | set |
| $\nu_u(f)$ | Consumption frequency of $f$ by $u$ | integer |
| $w_u(f)$ | Frequency weight $\nu_u(f)/\sum_{f'\in G_u}\nu_u(f')$ | $[0, 1]$ |
| $f^\star$ | Held-out target item in meal-level leave-one-out completion | element |
| **Losses and objectives** | | |
| $\mathcal{L}_{\text{retr}}$ | Retrieval loss (multi-positive InfoNCE over patients) | scalar |
| $\mathcal{L}_{\text{rank}}$ | Risk-aware ranking loss (pointwise/BPR/listwise) | scalar |
| $\mathcal{L}_{\text{aux}}$ | Auxiliary heads (e.g., clinical/text consistency) | scalar |
| $\mathcal{L}$ | Total loss | scalar |
| $\tau > 0$ | Temperature in InfoNCE (not the budget $\boldsymbol{\tau}_u$) | scalar |

# B  PATIENT PROFILE

We construct compact patient profiles by left-joining NHANES modules (1999–2023) on `SEQN`, mapping discrete codes to readable categories, and standardizing continuous variables with clinical cut-points. The numeric vector $\mathbf{c}_u$ parameterizes the nutrient budgets and safety weights per patient, while the 35–50 token text summary $\phi(u)$ is encoded to $\mathbf{t}_u$ for text-to-edge conditioning. Cycle provenance is retained for disambiguation and prefer fasting labs when flagged. Table 5 enumerates modules, key variables with examples.

## B.1  CODE-TO-PHRASE DICTIONARIES

Discrete codes are mapped to readable categories with small dictionaries that we release for reproducibility. Below shows an excerpt:

```
gender: {1: male, 2: female}
race:   {1: Mexican American, 2: Other Hispanic,
         3: Non-Hispanic White, 4: Non-Hispanic Black,
         6: Non-Hispanic Asian, 7: Other/Multi-Racial}
education: {1: <9th grade, 2: 9-11th, 3: HS/GED,
            4: some college/AA, 5: college graduate or above}
```

Table 5: NHANES modules for Patient Profile construction: modules, file prefixes, and representative variables.

| Module | File prefix | Key variables (examples) |
|--------|-------------|--------------------------|
| Demographics | DEMO_XX | RIDAGEYR (Age), RIAGENDR (Sex), RIDRETH3 (Race/Ethnicity), DMDEDUC2 (Education), DMDMARTL (Marital) |
| Examination | BMX_XX, BPX_XX | BMXBMI (BMI), BPXSY1 / BPXDI1 (SBP/DBP), additional valid BP repeats if available |
| Laboratory | GHB_XX, TCHOL_XX, BIOPRO_XX, GLU_XX | LBXGH (HbA1c), LBXTC (Total Cholesterol), LBXSCR (Serum Creatinine for eGFR), LBXGLU (Fasting Glucose) |
| Questionnaire | DIQ_XX, SMQ_XX | DIQ010 (Diabetes diagnosis), DIQ050 (Insulin use), SMQ020 (Smoking history) |
| Medications | RXQ_RX_XX | RXDDRUG (Drug name), RXDDRGID (Drug ID) |
| Dietary recalls | DR1TOT_XX, DR2TOT_XX | Day-level nutrient totals (e.g., added sugar, total carbs, fiber, sodium, energy), recall flags |

## B.2 Clinical Thresholds and Formulas

Continuous variables are standardized and categorized with widely used clinical cut-points. We cap extreme outliers by percentile clipping before scaling.

**BMI (kg/m$^2$).** *Underweight* <18.5, *Normal* 18.5–24.9, *Overweight* 25.0–29.9, *Obesity* ≥30.

**HbA1c (%).** *Normal* <5.7, *Prediabetes* 5.7–6.4, *Diabetes* ≥6.5.

**Blood pressure (mmHg; AHA 2017).** *Normal*: SBP<120 and DBP<80; *Elevated*: SBP 120–129 and DBP<80; *Stage 1 HTN*: SBP 130–139 or DBP 80–89; *Stage 2 HTN*: SBP ≥140 or DBP ≥90; *Crisis*: SBP ≥180 and/or DBP ≥120.

**Total cholesterol (mg/dL).** *Desirable* <200; *Borderline high* 200–239; *High* ≥240.

**eGFR (CKD-EPI 2021, race-free).** For serum creatinine Scr (mg/dL),

$$\text{eGFR} = 142 \cdot \min\left(\frac{\text{Scr}}{\kappa}, 1\right)^{\alpha} \cdot \max\left(\frac{\text{Scr}}{\kappa}, 1\right)^{-1.200} \cdot (0.9938)^{\text{Age}} \cdot \begin{cases} 1.012, & \text{if female} \\ 1, & \text{if male} \end{cases}$$

where $(\kappa, \alpha) = (0.7, -0.241)$ for females and $(0.9, -0.302)$ for males.

**CKD staging by eGFR (mL/min/1.73m$^2$).** G1: ≥90; G2: 60–89; G3a: 45–59; G3b: 30–44; G4: 15–29; G5: <15.

## B.3 Profile Template and Token Budget

We generate a compact text summary $\phi(u)$ (target 35–50 tokens) with optional clauses omitted if the source is missing:

```
{age}-year-old {race} {gender}, BMI {bmi} {bmi_cat},
HbA1c {a1c} {gly_cat}, eGFR {egfr} {ckd_cat},
BP {sbp}/{dbp} mmHg, TC {tc} mg/dL.
Education: {edu}; Marital: {marital}.
Diagnosed with diabetes: {diag}. Insulin use: {insulin}.
```

Priority for inclusion when truncating: clinical status (diabetes/HbA1c, eGFR/CKD) → cardio-metabolic vitals (BMI, BP, TC) → social descriptors (education, marital). We encode $\phi(u)$ with a domain language model to obtain $\mathbf{t}_u$.

## B.4 Numeric Composite and Heads

We assemble a standardized numeric vector

$$\mathbf{c}_u = \big[\text{age, sex, race, BMI, BP, HbA1c, eGFR, cholesterol, smoking, diabetes, insulin}, \dots\big],$$

apply z-scoring/min–max scaling with a learned affine imputer gated by a missingness mask $\mathbf{m}_u$. In our NHANES instantiation, dataset-specific identifiers are used only for tokenization or indexing and do not alter the graph structure.

## C  Nutrient Preprocessing and Embedding

For each food $f$, we construct a nutrient feature $\tilde{\mathbf{g}}(f)$ by winsorizing continuous nutrients at the 1st/99th percentiles, applying $\log(1 + x)$ to strictly positive entries, and z-scoring with train-set statistics. We then obtain a dense embedding

$$\mathbf{e}_f^{\text{nut}} = \text{LN}\big(\text{GELU}(W_g\, \tilde{\mathbf{g}}(f))\big) \in \mathbb{R}^{d_g},$$

and form the food node input as

$$\mathbf{x}_f = \big[\ \mathbf{e}_f^{\text{nut}}\ ;\ \mathbf{e}_f^{\text{cat}}\ ;\ \mathbf{t}_f^{\text{text}}\ \big],$$

where $\mathbf{e}_f^{\text{cat}}$ encodes categorical/group indicators and $\mathbf{t}_f^{\text{text}}$ is the short-description text embedding; numeric and text parts are fused via the FiLM gate used in the backbone. Note that $\mathbf{g}(f)$ in health constraints and penalties is kept in physical units, while $\tilde{\mathbf{g}}(f)$ is used only for representation learning.

# D  Patient Covariates Feature Engineering and the Patients-like-me Prior

## D.1  Variables (8 engineered features).

For each patient we construct an 8-dimensional clinical profile $\mathbf{x} \in \mathbb{R}^8$ from routinely available variables:

- **Age** (years), **BMI**, **A1c** (%);
- **Sex** (binary: male=1, female=0);
- **Diabetes status** (two dummy variables with "none" as the baseline: prediabetes, diabetes);
- **eGFR flags**: egfr_low = $\mathbb{K}\{\text{eGFR} < 60\}$, egfr_miss = $\mathbb{K}\{\text{eGFR is missing}\}$.

This yields 8 engineered features: 3 standardized continuous + 1 sex binary + 2 diabetes dummies + 2 eGFR flags. We use an 8-D *patient covariate* vector $\mathbf{x}_u \in \mathbb{R}^8$ comprising **demographic** (e.g., age, sex) and **clinical** markers (e.g., BMI, A1C). We refer to it as "patient covariates" rather than "clinical features" to avoid conflating demographics with clinical variables.

## D.2  Preprocessing.

Continuous variables (Age, BMI, A1c) are winsorized at the 1st/99th percentiles and $z$-scored per variable. Categorical fields are encoded as described above. Missingness in eGFR is modeled explicitly via egfr_miss; no forward imputation is used.

## D.3  distance and similarity.

Given two patients $u, v$, we define the standardized distance as the $\ell_2$ on the engineered vector:

$$d_{\text{clin}}(u,v) = \|\mathbf{x}_u - \mathbf{x}_v\|_2, \quad s_{\text{clin}}(u,v) = \exp\left(-\frac{d_{\text{clin}}(u,v)}{\sigma}\right),$$

where $\sigma$ is set to the median of $\{d_{\text{clin}}\}$ on the training population.

## D.4  User representation and Patient-like-me weights.

Let $\mathbf{z}_u \in \mathbb{R}^d$ be the learned user embedding from the PANACEA backbone with FiLM text modulation. We use cosine similarity for representation proximity, $s_{\text{rep}}(u,v) = \cos(\mathbf{z}_u, \mathbf{z}_v)$. To build *Patients-like-me* neighborhoods we combine clinical and representation similarity:

$$\tilde{s}(u,v) = s_{\text{rep}}(u,v) \cdot s_{\text{clin}}(u,v), \quad w_{uv} = \frac{\exp(\tilde{s}(u,v)/\tau)}{\sum_{v' \in \mathcal{N}_k(u)} \exp(\tilde{s}(u,v')/\tau)},$$

with temperature $\tau = 0.2$ and $k = 100$ nearest neighbors (by $\tilde{s}$). A hard gate on diabetes status (same stratum) can be applied before soft weighting.

## D.5  Neighborhood prior for retrieval.

From neighbors' training-window food logs we construct smoothed propensities

$$P(f \mid v) = \frac{c_{v,f} + \alpha}{\sum_{f'}(c_{v,f'} + \alpha)}, \quad \alpha \in \{0.5, 1.0\},$$

and aggregate a prior score

$$s_{\text{kNN}}(u,f) = \sum_{v \in \mathcal{N}_k(u)} w_{uv}\, P(f \mid v).$$

This prior is combined with user–food affinity at Stage-1:

$$s_{\text{stage1}}(u,f) = \cos(\mathbf{z}_u, \mathbf{z}_f) + s_{\text{kNN}}(u,f),$$

with per-user $z$-score normalization on each component before summation to avoid scale mismatch.

**Leakage guard and evaluation protocol.** All neighbor statistics $c_{v,f}$ and $P(f \mid v)$ use **training-window data only**. Evaluation uses the full catalog candidate pool; self-neighbors are removed; metrics are reported on users with non-empty test sets. We additionally report Kendall/Spearman correlation between $s_{\text{rep}}(u,v)$ and $s_{\text{clin}}(u,v)$ to quantify clinical alignment.

# E    TEXT-ONLY COLD-START BASELINES

**Goal.**    We quantify the *lower bound* of retrieval quality when using **text-only** semantic encoders under **strict entity-level cold-start**. This isolates how far text alone can go without any graph signal or patient covariates.

**Models.**    We use off-the-shelf **e5** and **SBERT** dual-encoders. The user/query and item towers share weights within each model.

**Input construction (text only).**    *Query text* concatenates light *meal context* tokens (e.g., meal type, time-of-day) into one sentence; *item text* is the food name plus a short nutrient phrase (e.g., kcal, carbs, sugar, sodium). **No** graph edges, numeric covariates, or hypergraph features are used.

**Training regimes.**    (i) **Zero-shot**: frozen encoders, cosine scoring. (ii) **Few-shot**: InfoNCE for **3–5 epochs** on train-only co-meal positives, in-batch negatives; **no** graph/covariates. Both regimes use identical tokenization and text templates.

**Cold-start protocol.**    Entity holdouts are applied *before* any training or indexing: (i) **Item-CS**: 10% foods never appear in train/val; test meals must contain these unseen foods. (ii) **User-CS**: 20% users fully held out. (iii) **Hybrid-CS**: intersection of the two. Retrieval uses **ANN (FAISS)** over the full candidate set with cosine similarity; we evaluate **Recall@50** and **NDCG@50** (percentages).

**Leakage controls.**    Positives/negatives and all text corpora are built *only* from the training split; no statistics or edges from val/test are used.

**Takeaways.**    Across all splits, text-only dual encoders stay around **1%** Recall@50 and $\leq$**0.24%** NDCG@50, and they *collapse* in Hybrid-CS. This establishes a clear *lower bound* and motivates the main approach that combines **hypergraph structure** (meals-as-hyperedges) with **patient-aware** conditioning (node-wise FiLM). In the main paper, structural baselines (LightGCN/HGNN) and **Ours** are evaluated under the same cold-start protocol for head-to-head comparison.

# F    USE OF LLMS

We used LLMs solely for writing assistance during the preparation of this manuscript. Specifically, we employed LLM-based tools (ChatGPT, Claude) to assist with: (1) proofreading and grammar checking of manuscript drafts; (2) improving clarity and readability of technical descriptions; (3) generating alternative phrasings for complex concepts. All technical content, methodology, experimental design, results, and scientific claims were developed independently by the authors without LLM assistance.

