# OpenReview forum: "PanaCea: Clinical Hypergraph Framework for Health-Aware Personalized Diet Recommendation"
_ICLR.cc/2026/Conference — Submitted to ICLR 2026_

### Official Review · Reviewer_qTYX · 2025-10-14

**Soundness:** 3
**Presentation:** 3
**Contribution:** 1
**Rating:** 2
**Confidence:** 4

**Summary:**

The paper proposes PanaCea, a clinical hypergraph framework for personalized and health-aware diet recommendation. Unlike traditional recommenders that treat foods as independent items, PanaCea models each meal as a hyperedge connecting a patient, the meal context (type, time, location), and multiple co-consumed foods. The system integrates three authoritative datasets—NHANES dietary and health records, FNDDS nutrient composition, and Food Compass healthfulness scores—to create clinically grounded patient and food profiles. The framework operates in two stages: (1) a hypergraph-based multi-task retrieval module that aligns patient preferences, clinical similarity, and meal completion signals, and (2) a nutrition-aware re-ranking stage that applies patient-specific safety constraints (e.g., sodium, sugar, glycemic risks). Evaluated on NHANES, PanaCea outperforms collaborative filtering and graph-based baselines on both ranking metrics and nutrition-focused measures, improving constraint satisfaction rates from 61% to 74%. The work is the first to embed clinical safety directly into the learning objective for dietary recommendation, offering a scalable and adaptable pipeline for diverse health conditions.

**Strengths:**

it's a good and important topic, I encourage the authors to further work on this domain.

**Weaknesses:**

First, I believe this is a work more suitable for data mining rather than machine learning, as it seems to me the contributions lie in the curation of a very specific domain dataset of a very specific domain topic. And the method, is neither novel or generalizable. All experiments are done on this dataset without discussion of any other public benchmarks.

Second, disregard of the method novelty, the data mining work is also very poor. The authors ignored numerous prior works using NHANES data for "Health-aware Personalized Diet Recommendation" [1-3] or potentially other more generalized health-aware FRS benchmarks [4]. The proposed method, though using hyper-graph, but generally followed the same data processing logics compared with prior works, e.g., NGQA [2], but the authors considered even fewer and less comprehensive factors. Without proper literature review, this work naturally lacks proper discussion of the motivations of why creating new benchmarks or why using hypergraphs.

Third, the experiments are extremely limited. It basically has no other baselines, the ablations are also limited.  In sum, this work is clearly below the bar of ICLR.

[1] Zhang, et al. Mopi-hfrs: A multi-objective personalized health-aware food recommendation system with llm-enhanced interpretation. KDD. 2025.
[2] Zhang, et al. NGQA: a nutritional graph question answering benchmark for personalized health-aware nutritional reasoning. ACL. 2025
[3] Shi, et al. NG-Router: Graph-Supervised Multi-Agent Collaboration for Nutrition Question Answering. Arxiv. 2025
[4] Bölz, et al. Hummus: A linked, healthiness-aware, user-centered and argument-enabling recipe data set for recommendation. RecSys. 2023

**Questions:**

The authors introduced "multi-modal fusion" in Figure 1, and marked it in the keywords. But I didn't see detailed discussion about fusing data from other modality. Did the authors consider the encoding text into a hypergraph multi-modal? Or are there other data modalities I didn't notice?

---

### Official Review · Reviewer_f8gm · 2025-10-23

**Soundness:** 2
**Presentation:** 2
**Contribution:** 2
**Rating:** 2
**Confidence:** 5

**Summary:**

PanaCea proposes a hypergraph neural network framework for personalized, health-aware food recommendations. The core innovation is modeling each meal as a hyperedge connecting one patient to multiple co-consumed foods, capturing multi-way nutritional interactions that pairwise graphs cannot represent. The system integrates three data sources (NHANES dietary records, FNDDS nutritional composition, Food Compass 2.0 quality scores) and employs a two-stage architecture: Stage 1 performs multi-task learning on user-item preference, clinical patient similarity, and meal completion; Stage 2 applies nutrition-aware re-ranking with patient-specific soft constraints. Evaluation on NHANES demonstrates improvements in both ranking metrics and nutrition-focused measures (constraint satisfaction rate, nutritional quality).

**Strengths:**

1. **Well-motivated problem**: Diet-related diseases are a major public health challenge. Personalized, clinically-grounded recommendations are needed.

2. **Thoughtful data integration**: Combining population-scale dietary patterns (NHANES), authoritative nutrient databases (FNDDS), and evidence-based quality scores (Food Compass 2.0) is comprehensive and principled.

3. **Clinical awareness**: Unlike many recommenders, PanaCea attempts to encode patient-specific health conditions (diabetes status, HbA1c, eGFR, BP) into the recommendation objective rather than as post-hoc filtering.

4. **Reasonable representation choice**: Modeling meals as hyperedges is intuitive for capturing multi-way food interactions—a real limitation of pairwise graph approaches.

5. **Multiple evaluation protocols**: Both patient-union and meal-LOO protocols; both traditional (nDCG, recall) and nutrition-specific metrics (CSR, MFCS) are thoughtful.

6. **Reproducibility commitment**: Promise to release code and detailed documentation (Appendices B-D) aids reproducibility.

7. **Ethics statement**: Authors acknowledge limitations and position PanaCea as a decision-support tool, not a replacement for medical advice.

**Weaknesses:**

1. **No clinical validation of recommendations**:
   - Evaluation is entirely on historical consumption patterns (NHANES). No evidence that recommendations are clinically appropriate.
   - Ground truth (past consumption) may encode unhealthy choices. Training on this could perpetuate dietary harm.
   - No prospective validation, clinical outcome measurement, or expert (registered dietitian) review of recommendations.

2. **Weak experimental design**:
   - Missing comparisons with cited health-aware systems (FoodKG, MealRec+, DFRS, etc.)
   - Only collaborative (LightGCN) and hypergraph (HGNN) baselines—no clinical adaptation baselines
   - Text-only cold-start baselines (Table 1) are not comparable to the full system
   - No significance testing, confidence intervals, or effect size reporting

3. **Technical clarity and justification gaps**:
   - Multi-task loss scheduling ("ramps λ_pl, λ_meal") not rigorously defined; no sensitivity analysis
   - Clinical projection ψ_clin: Is it learned? How? Where in the architecture?
   - Stage-2 re-ranking (Eq. 9) is simplistic: multiplicative penalty + soft constraint. Why this form? No justification for design choices.
   - Hard negative selection procedure not specified, yet this significantly affects contrastive learning
   - "norm(·)" in Eq. 9 not formally defined

4. **Evaluation concerns**:
   - **Meal-LOO may not reflect real recommendation scenarios**: Holding out items from observed meals allows the model to learn co-consumption patterns from training data. Real recommendations must construct entire meals from scratch or small contexts.
   - **No temporal generalization test**: Does the model generalize to future eating patterns outside the training temporal window?
   - **Retrospective only**: Matching against past consumption is not validation that recommendations improve health

5. **Missing practical considerations**:
   - Allergies and intolerances not handled (need hard constraints, not soft)
   - Medications not incorporated (mentioned as future work; critical for safety)
   - Cost, availability, cultural preferences not modeled
   - The system cannot enforce true dietary safety for conditions like celiac disease or severe allergies

6. **Generalization limitations**:
   - U.S.-only evaluation; Food Compass 2.0 may not transfer internationally
   - NHANES population may not represent those with poorest diet quality or greatest medical need
   - Acknowledged as future work, but limits current impact

7. **Incomplete ablations**:
   - No ablation on individual multi-task components (Eq. 4, 5, 6)
   - No ablation on clinical projection ψ_clin (Table 3 only tests without covariates, not the projection itself)
   - No sensitivity analysis for temperature parameters τ_1, τ_2, τ_3 or neighborhood size k

8. **Reproducibility gaps**:
   - Many hyperparameters unspecified or underspecified
   - "Warm-up then ramps" training procedure vague
   - Per-item nutrition risk r_nut(u, f) mapping described as "smooth" but functional form not given
   - Patient profile truncation procedure (B.3) involves priority ordering—edge case behavior unclear

9. **Limited technical novelty**:
   - Hypergraph neural networks are established (cited: Feng et al. 2019, Chien et al. 2022, Xia et al. 2021/2022)
   - Multi-task learning with InfoNCE is standard (ESMM, MMoE, PLE cited but not adapted specifically)
   - Clinical projection and soft constraints are reasonable but not deeply novel

10. **Concerning use of past consumption as ground truth**:
   - If a patient with poor dietary habits (high sodium, sugar intake) is in the training data, the model learns those patterns are "good" recommendations
   - This could reinforce harmful eating patterns rather than nudge toward healthier choices
   - Using past consumption as ground truth assumes consumption history = appropriate dietary choices, which is not true in a clinical setting

**Questions:**

## Questions for Authors

1. **Clinical validation**: Can you provide any post-hoc validation against clinical guidelines (e.g., ADA diabetes guidelines, DASH diet for hypertension) or expert review from registered dietitians? This is essential for a clinical recommendation system.

2. **Baselines**: Why were FoodKG and MealRec+ not included in experiments? They are cited as relevant work but not compared. Can you provide results?

3. **Ground truth appropriateness**: How do you justify using historical consumption as ground truth when patients' past eating patterns may be clinically suboptimal? Have you validated that recommended foods are indeed more appropriate than held-out items?

4. **Multi-task loss schedule**: Please provide the exact schedule for λ_retr, λ_pl, λ_meal during training. Include sensitivity analysis.

5. **Clinical projection**: Is ψ_clin learned end-to-end or fixed? If learned, provide ablation results with and without it.

6. **Hard constraints vs. soft**: For severe conditions (allergies, celiac), soft constraints are inappropriate. How does the system handle these?

7. **Meal-LOO generalization**: In real deployment, recommendations must construct meals from scratch. How does the model perform when constructing complete meals for new eating contexts (e.g., a restaurant meal, not seen in training)?

8. **Statistical rigor**: Can you provide confidence intervals, p-values, or significance tests for the main results?

9. **Hyperparameter sensitivity**: Provide ablations on τ_1, τ_2, τ_3, k (neighbor size), and α (smoothing in propensity).

10. **Food Compass validity**: Food Compass 2.0 is relatively recent (2024). Has it been independently validated in dietary intervention studies?

**Details Of Ethics Concerns:**

- ☑ **Privacy, security and safety**: Health data from NHANES is de-identified but highly sensitive. Deployment could expose individual dietary patterns.
- ☑ **Potentially harmful insights, methodologies and applications**: A system that recommends foods based on past consumption could reinforce unhealthy eating in patients with poor diets. Without clinical validation, recommendations could be harmful.
- ☑ **Responsible research practice**: Clinical recommendation systems should undergo more rigorous validation (clinical trials, expert review) before deployment.

**Specific concerns:**

1. **Harm from biased ground truth**: Using NHANES consumption as ground truth can perpetuate and reinforce unhealthy eating patterns if applied to patient recommendations. Patients with dietary risk factors may receive recommendations aligned with their past poor choices, not clinically optimal alternatives.

2. **Lack of clinical oversight**: The paper positions this as a decision-support tool but provides no evidence of safety or clinical appropriateness. Deploying without clinical validation could cause harm.

3. **False authority**: A neural network's recommendations may be perceived as clinically authoritative even without validation, leading patients to trust unsafe advice.

4. **Regulatory compliance**: If deployed in a clinical setting, such a system may require FDA clearance as a medical device. Current evidence is insufficient for that.

**Mitigation recommendations:**
- Obtain expert review from registered dietitians for a sample of recommendations
- Validate against established clinical dietary guidelines
- Implement clear disclaimers and human-in-the-loop oversight
- Conduct prospective validation before any clinical deployment
- Address privacy safeguards explicitly if scaling to real patient populations.

---

### Official Review · Reviewer_5fgL · 2025-11-01

**Soundness:** 2
**Presentation:** 1
**Contribution:** 3
**Rating:** 4
**Confidence:** 3

**Summary:**

This paper introduces PANACEA, a novel hypergraph-based framework for personalized, health-aware food recommendation. The authors address critical flaws in existing systems, which often ignore the complex interactions of foods within a meal and fail to account for patient-specific health conditions.
The core contribution is the representation of each eating occasion as a "meal-as-set" hyperedge, which connects a patient node to the set of co-consumed food nodes. This allows the model to capture higher-order interactions that standard graph models miss.

**Strengths:**

- The paper's primary strength is the insight to model meals as hyperedges. This is a natural and powerful way to capture the set-based nature of eating, where nutrient effects are cumulative and interactive. The experimental gains of HGNN over LightGCN (Table 2) provide clear evidence for the value of this higher-order structural information.
- The framework does not just filter by clinical rules post-hoc; it injects clinical awareness directly into the learning process. The multi-task objective's clinical alignment task ($\mathcal{L}_{pl}$) and the "patients-like-me" kNN prior (Eq. 8) are intelligent ways to force the embedding space to respect clinical similarity. The reported negative correlation ($\rho = -0.329$) between embedding similarity and clinical distance is strong evidence that this alignment is successful.
- The fusion of three distinct, large-scale, and authoritative datasets (NHANES, FNDDS, Food Compass 2.0) is a massive data engineering effort that provides an unprecedentedly rich foundation for this problem. This is a significant contribution to the community, especially with the promise to release the processing pipelines.
- The introduction of nutrition-centric indices like Constraint Satisfaction Rate (CSR) and Mean Fractional Constraint Satisfaction (MFCS) is essential for evaluating this task. The results in Table 3—where LINeR (Stage-2) dramatically improves CSR from 0.6120 to 0.7435 while maintaining relevance—are very convincing and directly validate the two-stage design.

**Weaknesses:**

1. The LINeR re-ranking stage, while effective, is the least-developed part of the paper. It is described as a non-trained module that applies "diabetes-focused continuous penalty" $c_{dm}(u,f)$. This raises several concerns:
- How were the risk functions $r_{nut}$ and $c_{dm}$ (Eq. 9) designed? The paper mentions "hyperparameters... are validated", but this sounds like manual tuning. More explanation or illustrative examples are needed to clarify how these hyperparameters were defined and validated.
- The current design seems hard-coded for diabetes. It is unclear how the system would handle patients with complex, comorbid conditions such as chronic kidney disease (CKD) and hypertension, which impose different or even conflicting nutritional constraints (e.g., on protein, sodium, or potassium intake). In practice, diabetic patients often have multiple co-existing conditions. A truly “clinically-aware” framework should generalize beyond a single disease—ideally by learning a re-ranking policy rather than relying on hand-crafted penalties for each condition.

2. The framework is evaluated on over 20 years of NHANES data, which should result in an extremely large hypergraph. However, the paper omits fundamental statistics such as the number of patients ($|\mathcal{U}|$), foods ($|\mathcal{F}|$), and meal hyperedges ($|\mathcal{E}|$). There is also no discussion of computational complexity, training time, or inference latency. In particular, the “patients-like-me” kNN prior (Eq. 8) could pose a significant inference bottleneck. Furthermore, key statistics of the three datasets and descriptive hypergraph properties are missing, limiting reproducibility and scalability assessment.

3. The paper repeatedly mentions the importance of context (meal type, time, location). Figure 3 even shows a "Context Node". However, it is unclear how this context is formally used in the HGNN backbone (Eq. 1) or the multi-task losses (Eq. 4, 5, 6). The hyperedge attribute matrix $A_e$ is mentioned but does not appear in the formulations. Is this context information only used in the Stage-2 $\phi_{meal}$ term, which itself is not described?

4. Formatting Issues, such as overlapping of  Equation 3 and its accompanying text.; review and adjust line spacing throughout the paper—some lines (e.g., 208/209, 322/323) appear overly compressed, while others are unevenly spaced.

**Questions:**

1. Could the authors please elaborate on the formulation of the risk penalties $r_{nut}$ and $c_{dm}$ in Eq. 9? How can this re-ranking framework be generalized to support patients with multiple, co-morbid health conditions (e.g., diabetes + CKD) without requiring a new, hand-crafted penalty function for each combination?

2. Explaining the result in Table 2 where your Stage-1 model achieves state-of-the-art Recall@10 but has a substantially worse nDCG@10 than the baselines as well as Table 3 full model still underperforms baselines? Why does the model struggle with ranking in the top positions for this task?

3. Figure 2 is visually striking, but its interpretation is unclear. The legend defines nutrient axes, but what is the practical insight from the "meal curves" and "tethers"? Does the shape or volume of the meal surface have a quantitative meaning, or is this purely an illustrative visualization?

---

### Official Review · Reviewer_wi28 · 2025-11-02

**Soundness:** 1
**Presentation:** 2
**Contribution:** 1
**Rating:** 2
**Confidence:** 2

**Summary:**

The paper proposes PanaCea, a two-stage health-aware food recommendation framework that models each eating occasion as a meal-as-set hypergraph linking a patient, contextual attributes, and co-consumed foods. Stage-1 performs multi-task retrieval over a hypergraph neural backbone with three contrastive objectives: user-to-item preference, "patients-like-me" clinical alignment, and set-to-item meal completion. Stage-2 applies nutrition-aware re-ranking with soft, patient-specific penalties and optional disease-focused constraints. The system is instantiated on U.S. datasets by integrating NHANES dietary logs, FNDDS nutrient composition, and Food Compass-2.0 quality scores. Evaluation uses two protocols, Patient-Union (user-level recall) and Meal-LOO (within-meal completion), reporting ranking metrics alongside nutrition-centric indices such as constraint satisfaction rate. Reported results show gains over LightGCN and a basic HGNN backbone, and an increase in constraint satisfaction after Stage-2 re-ranking.

**Strengths:**

1) Problem framing and representation. Modeling meals as hyperedges is a natural and defensible way to capture multi-way nutrient interactions and co-consumption patterns that pairwise graphs elide. The "patients-like-me" prior is a principled way to fuse collaborative signals with clinically grounded similarity.

2) Clinically motivated objectives. Elevating clinical safety to a first-class objective, rather than a post-hoc filter, is an important step for health-aware recommendation. The proposed nutrition-aware re-ranking surfaces an actionable pathway to condition slates on patient covariates and disease status.

3) Multi-source data integration. The alignment of NHANES, FNDDS, and Food Compass is non-trivial and has practical value. The paper describes a reusable pipeline that many in the community could adapt to similar settings.

**Weaknesses:**

1) Baselines are limited for the setting. The comparison focuses on LightGCN and a hypergraph backbone without modulation. Modern, strong session or set-aware baselines are missing or only cited (e.g., HCCF, DHCN, HGAT, SGL, SASRec, GRU4Rec). Without these, the claimed gains may be overstated.

3) Metric reporting and statistical rigor. Several tables present single-point improvements without confidence intervals, statistical tests, or per-stratum breakdowns (e.g., diabetes vs. non-diabetes, CKD stages). Given NHANES heterogeneity, stratified performance and significance testing are needed to claim reliability.

3) Text and figure overlap issues. Several lines in the main text show overlapping issues, particularly in Stage-1, which affect readability.

**Questions:**

1) How exactly is the clinical projection psi_clin constructed? Is it trained end-to-end, or fixed from clinical embeddings? What regularization strategies prevent overfitting to sparse patient covariates?

2) How does the hypergraph model handle rare or unseen foods with sparse connections? Are inductive encoders supported for new items entering the system?

3) Why are state-of-the-art set-aware and session-based models absent from the comparison (e.g., HCCF, DHCN, HGAT, SASRec)? Have these been evaluated internally, and if so, what were the results?

4) Several presented improvements lack confidence intervals or significance testing. Can the authors provide statistical analyses, preferably stratified by disease categories or demographic subgroups?

---

### Meta-Review · Area_Chair_cFrE · 2026-01-09

**Summary:**

This paper proposes a hypergraph-based framework for health-aware food recommendation built on a curated NHANES-based dataset. Reviewers agree that the contribution of this work is limited, as the primary contribution appears to be dataset curation within a narrow application domain rather than a novel or generalizable machine learning method. The proposed modeling approach is neither technically novel nor demonstrated to be broadly applicable, and all experiments are limited to a single, self-constructed dataset with no evaluation on established public benchmarks. In addition, the paper does not adequately engage with prior work. Several relevant studies using NHANES and related datasets for health-aware personalized diet recommendation and food recommendation systems are omitted, and the relationship between this work and existing methods (e.g., those with similar data processing pipelines) is not properly discussed. No rebuttal was provided to address these concerns. Taken together, we recommend rejecting this work.

**Reviewer Concerns:**

No response provided by the authors.

**Reviewer Scores:**

No response provided by the authors.

---

### Decision · Program_Chairs · 2026-01-26

Reject